# Use of H_2_O_2_ to Cause Oxidative Stress, the Catalase Issue

**DOI:** 10.3390/ijms21239149

**Published:** 2020-11-30

**Authors:** Céline Ransy, Clément Vaz, Anne Lombès, Frédéric Bouillaud

**Affiliations:** Institut Cochin, Université de Paris, INSERM, CNRS, F-75014 Paris, France; celine.ransy@inserm.fr (C.R.); clement.vaz@inserm.fr (C.V.); anne.lombes@free.fr (A.L.)

**Keywords:** reactive oxygen species, oxidative damage, DNA strand break, cellular respiration, aconitase, fumarase

## Abstract

Addition of hydrogen peroxide (H_2_O_2_) is a method commonly used to trigger cellular oxidative stress. However, the doses used (often hundreds of micromolar) are disproportionally high with regard to physiological oxygen concentration (low micromolar). In this study using polarographic measurement of oxygen concentration in cellular suspensions we show that H_2_O_2_ addition results in O_2_ release as expected from catalase reaction. This reaction is fast enough to, within seconds, decrease drastically H_2_O_2_ concentration and to annihilate it within a few minutes. Firstly, this is likely to explain why recording of oxidative damage requires the high concentrations found in the literature. Secondly, it illustrates the potency of intracellular antioxidant (H_2_O_2_) defense. Thirdly, it complicates the interpretation of experiments as subsequent observations might result from high/transient H_2_O_2_ exposure and/or from the diverse possible consequences of the O_2_ release.

## 1. Introduction

Exposure to hydrogen peroxide (H_2_O_2_) is a widely used procedure to cause oxidative damage/stress in cellular models. The Fenton’s reaction between H_2_O_2_ and Fe^2+^ ions generates the highly reactive OH radical and is thought to be the main mechanism for oxidative damage [1]. Then the question arises as to whether the conditions used are relevant to the mechanism underlying endogenous oxidative damage. Oxidative damage to cells is supposed to result primarily from oxygen using reactions within cells: cellular respiration accounts for most of the cellular oxygen consumption hence mitochondria are supposed to be oxidative stress generators. A quick survey of the literature reveals that the concentrations of H_2_O_2_ used to trigger oxidative damage appear disproportionately high to anyone used to dealing with cellular respiration for the following reasons: Dissociation of oxygen from hemoglobin starts below 50 µM and interstitial concentration is in the 20 µM range [2] hence even lower within cells. The affinity of mitochondria for oxygen is extremely high [3,4] and consequently mitochondria can exhaust all incoming oxygen with minimal decrease in their respiratory rate. This allows a sustained oxygen consumption flux while oxygen concentration is close to zero [5]. This points to the distinction to be made between concentrations and fluxes. Several questions arise with regard to this H_2_O_2_ model of oxidative damage: (1) Why are these high concentrations of H_2_O_2_ necessary to observe oxidative damage? (2) Are there other consequences than direct oxidative damage by H_2_O_2_? (3) Are the induced modifications relevant to physiology and to the redox balance associated with aerobic life?

The aim of the present report is to attract attention to the fact that direct application of H_2_O_2_ to cells results in the generation of oxygen (O_2_) because of the presence of catalase [6,7,8]. This is not unexpected but seems generally overlooked and is likely relevant to the above first two questions. With regard to the third question, which lies well beyond the scope of the present article, two comments arise: (1) High concentration of H_2_O_2_ may be justified with the belief that they would increase the frequency of oxidative damages but would not alter their nature. (2) In addition to normal aerobic metabolism, local concentrations might be closer to the experimental values mentioned before in the inflammatory state where enzymatic reactions generate oxygen radicals with the aim to kill unwanted organisms/cells. 

Actually, this report originates from our attempts to generate DNA strand breaks in mitochondrial DNA with hydrogen sulfide [9,10]. In our hands hydrogen sulfide could neither cause nuclear [11] nor mitochondrial DNA damage, while H_2_O_2_ as a positive control required the high micromolar concentrations found in the literature (Appendix A). 

Lipids and proteins are other targets for oxidative stress. The enzyme aconitase contains an iron atom (Fe^2+^) sensitive to oxidative stress [12]. Its mitochondrial isoform is part of the tricarboxylic acid cycle (Krebs cycle). In contrast, fumarase, whose mitochondrial isoform is also part of the Krebs cycle, is resistant to oxidative stress. Similar experimental procedures detect aconitase and fumarase activities allowing sequential monitoring of these two reactions in the same reaction medium. Therefore, the aconitase to fumarase ratio is a convenient index of the extent of oxidative damages/stress [12]. Examination of this ratio in cells exposed to H_2_O_2_ could not detect indication of intracellular oxidative damage to aconitase unless H_2_O_2_ approached the millimolar concentration. 

## 2. Results

### 2.1. Values for H_2_O_2_ Concentration in the Literature

The PubMed database was searched with the keywords «Oxidative damage and Hydrogen Peroxide» and only publications in open access were considered. This request yielded 5147 publications. They were sorted according to best match. The first 201 were scrutinized for H_2_O_2_ treatment with concentration values, yielding 112 publications, whose PMID identifiers are listed in Appendix B. Different experimental profiles had to be considered: 46 publications used a single H_2_O_2_ concentration value, 36 publications presented a dose response curve leading to the choice of a single concentration used throughout the rest of the study (this concentration was considered here), and 30 publications used a set of increasing concentrations with no privileged concentration leading us to consider the mean value between minimal and maximal concentrations. Then the repartition of the H_2_O_2_ concentrations used was as follow: <10 µM 4 publications, 10–99 µM 14 publications, 100–500 µM 55 publications, 501–1000 µM 13 publications, and >1000 µM 26 publications. The concentrations of H_2_O_2_ used to generate oxidative damage in cellular models are therefore in the high micromolar range with 84% above 100 µM. Notably the 100–500 µM range was used in half of the publications.

### 2.2. A Proportionate Increase in Dioxygen Follows High Micromolar H_2_O_2_ Addition

Our initial aim was to evaluate mitochondrial DNA strand break in conditions of induction of mitochondrial sulfide oxidation in wild type chinese hamster ovary (CHO) cells or in cells with overexpressed human sulfide quinone reductase [13]. We used the O2k oxygraph, which allows simultaneous measurement in two identical chambers (Figure 1). It showed that addition of high (hundreds) micromolar H_2_O_2_ resulted in an immediate release of molecular oxygen apparently proportionate to the H_2_O_2_ input, and 500 µM H_2_O_2_ led to oxygen concentrations above 250 µM, hence higher than that resulting from air saturation of the medium (approx. 200 µM at 37 °C). However, with 25 µM H_2_O_2_ the oxygen release was hardly detected.

We hypothesized that catalase enzymatic activity (H_2_O_2_ H_2_O + ½ O_2_) underlay the observed phenomenon. The reaction equation predicts that the oxygen concentration increase would equal half the concentration of H_2_O_2_ added, hence 12.5, 50, and 250 µM in the examples shown in Figure 1. Figure 2 shows the percentage of this theoretical oxygen increase that was actually measured in the experiments shown in Figure 1. More than 80% of predicted oxygen was recovered with 500 µM H_2_O_2_, roughly 50% with 100 µM, and this percentage dropped below 10% with 25 µM H_2_O_2_, at which the slow linear increase in the long term (120 s and more) is rather explained by small experimental differences exaggerated by the conversion in % of absolute values differing by two orders of magnitude. When the difference between two simultaneous experiments shown in Figure 1 was considered (empty circles in Figure 3), the percentage of O_2_ recovery was higher and reached 100% when the difference between 500 µM and 100 µM traces (hence 400 µM H_2_O_2_) was considered. 

Putting together the results of all the experiments made with H_2_O_2_ additions in the 25–500 µM range and with CHO cells suspensions close to 2 × 10e6 cells/mL suggested a simple model (Figure 3). In that model two reactions took place: an immediate titration of a part of H_2_O_2_ and catalase reaction on the rest of H_2_O_2_. With the lowest H_2_O_2_ concentration (25 µM) the immediate titration checked almost 100% of the H_2_O_2_ and little was made available to catalase (Figure 1 and Figure 2). 

Analysis of a range of cell concentrations strengthened the proposed role of cellular catalase (Figure 4). With increasing cell concentration, the lag time between H_2_O_2_ addition and maximal O_2_ release decreased (Figure 4a). A single experiment with medium alone led to a low rate of O_2_ release that reached a peak of 27 µM, hence 11% of the value explained by a catalase like reaction, after 1518 s (not shown on Figure 5). Altogether, this meant that the more cells, the faster the catalase reaction.

However, when the yield of the catalase reaction was considered it declined when lower amounts of cells were used. Time needed for completion of catalase reaction was then likely to be the determinant factor. To a significant extent experimental limitation could explain this because the determination of time and extent of the maximal O_2_ release became increasingly inaccurate when it took a long time. Another proposal would be that in addition to fast titration (see above) other slower reactions, which could involve components of the culture medium or renewal of H_2_O_2_ quenchers, contributed to H_2_O_2_ elimination. 

### 2.3. Fast Rates of Catalase Reaction 

Within three minutes, 2 × 10^6^ CHO cells neutralized 500 µM H_2_O_2_ (Figure 4a). This corresponds to an average rate while the initial rate was much faster (Figure 1). We estimated this initial rate and Figure 5a represents its dependence on H_2_O_2_ concentration with CHO cells. This initial rate of oxygen release remained proportionate to H_2_O_2_ concentration in the range 0–1000 µM with no saturation. The oxygen release rate could largely outperform the cellular oxygen consumption rate (cellular respiration) with rates that are more than one order of magnitude higher in the opposite direction (Figure 5a). This fits with the very high turnover rate of the catalase catalytic cycle [6] as a consequence while minutes were needed to reach the maximal increase in O_2_ (Figure 4a), hence to exhaust all H_2_O_2_ in the external medium the H_2_O_2_ concentration decreased sharply within seconds after being in contact with cells.

We observed comparable rates for this catalase reaction in eight cell lines. Interestingly, the O_2_ release rate (catalase reaction) apparently correlated with the endogenous respiration rate, i.e., the cellular oxidative metabolism (Figure 5b).

### 2.4. Cellular Respiration and Aconitase Activity Resistant to H_2_O_2_ in the High Micromolar Range

We evaluated intracellular oxidative stress using the aconitase to fumarase ratio, which is considered a sensitive index [12]. Furthermore, we discriminated the cytosolic enzymes, revealed after mild digitonin treatment of the cells, from the mitochondrial enzymes, requiring drastic disruption of the mitochondrial inner membrane using Triton X100 (Figure 6). Increasing concentrations of H_2_O_2_ up to 500 µM did not alter the aconitase to fumarase ratio, either in cytosol or in mitochondria (Figure 6a). One experiment with H_2_O_2_ concentration raised up to 1mM induced a detectable decrease in cytosolic aconitase activity that remained partial for the mitochondrial enzyme (Figure 6b). 

Evaluation of a potential impact of H_2_O_2_ impact on cellular oxygen consuming reactions was impossible in the short term after H_2_O_2_ addition because of the O_2_ release. However, with the rapid decline of H_2_O_2_ concentration, cellular oxygen consumption rate became detectable, gradually converging towards a new stable value. Comparison of that value with the value observed just before H_2_O_2_ addition, used as reference, did not show any change with H_2_O_2_ ranging from 0 to 750 µM (Figure 7 and Figure 8). This was true within the period considered, limited to 30–40 min after H_2_O_2_ addition. Respiration showed a mild increase (>110%) with 1mM H_2_O_2_, likely explained by the huge increase in O_2_ concentration up to three times the level for saturation with air (not shown). With the reservation that we could not distinguish between mitochondrial respiration and other oxygen consuming processes, this suggested that intense exposure to H_2_O_2_ and subsequently to oxygen did not result in fast and irreversible damage to any critical component necessary for normal cellular respiration. This was fully consistent with the resistance of mitochondrial aconitase to the exogenous H_2_O_2_. 

## 3. Discussion

The results presented here indicate that cells subjected to H_2_O_2_ exposure convert H_2_O_2_ into O_2_ within a few minutes. This makes the H_2_O_2_ exposure considerably shorter and less intense than expected. It may well explain why such high H_2_O_2_ concentrations were required to record detectable damages. The conversion of hundreds of micromolar H_2_O_2_ into oxygen has the consequence of exposing cells to O_2_ concentrations higher than that resulting from equilibration of the aqueous medium with air, a concentration already way above physiological oxygen concentrations [2]. 

The oxidative damages taking place after H_2_O_2_ exposure are thought to result for a significant (largest) part from the hydroxyl radical liberated by Fenton’s reaction. However, a surge in superoxide has been associated with exposure to H_2_O_2_ [14]. Superoxide results from O_2_ reduction with a single electron, and is converted into H_2_O_2_ by superoxide dismutase. Occurrence of the reverse reaction, with generation of superoxide from H_2_O_2_, is not immediately obvious and is expected to be an indirect consequence of H_2_O_2_ exposure [15]. Our experiments indicate that a large increase in oxygen concentration is expected to take place at the site of catalase action hence inside the cell. This rise in intracellular O_2_ would greatly increase the probability of its reduction by leakage of a single electron from cellular metabolism. It may therefore provide a direct explanation for the observed increase in cellular superoxide production. 

In our experiments, the equilibration of concentrations was immediate because of the stirring of the cellular suspension necessary for the measurement of cellular oxygen consumption. Consequently, cellular catalase could access added H_2_O_2_ within a short time. However, in cells attached at the bottom of a well and surrounded by a still medium, O_2_ and H_2_O_2_ may have restricted diffusion leading to delayed access to catalase. H_2_O_2_ may then represent an additional oxygen reserve. Interestingly, the dose response curve showed a rather positive effect of H_2_O_2_ at low concentrations, before the deleterious effect took place/dominated [16]. Reactive oxygen species (ROS) signaling might explain this observation but the improvement of cellular bioenergetics by H_2_O_2_/catalase mediated O_2_ supply to mitochondria may also deserve consideration. 

At low concentrations of H_2_O_2_, a significant proportion escaped from catalase action (Figure 2 and Figure 3). One could easily envision that H_2_O_2_ would first titrate all the molecules able to react quickly with it. They would represent a “sink for H_2_O_2_”, which includes components of the medium as well as cellular antioxidant defenses. This sink shows highest affinity but limited size (40 nanomoles in our experimental settings). In contrast, catalase shows relatively low affinity but a large activity able to neutralize within a few minutes surrounding H_2_O_2_ concentrations approaching the millimolar range, hence a micromole of H_2_O_2_ in our settings. 

Altogether these observations render the interpretation of H_2_O_2_ experiments far more complex than considering a sudden increase in the probability for Fenton’s reaction to occur (Figure 9).

ROS release by mitochondria often leads to considering them as permanent cell threatening ROS generators, checked by antioxidant defenses. Decline of the latter would explain increase in oxidative damage. Mitochondrial superoxide dismutase (MnSOD) quickly converts mitochondrial superoxide, the essential mitochondrial ROS, into H_2_O_2_. Then, if the largest part of cellular oxidative damage has a mitochondrial origin, the mitochondrial exposure to H_2_O_2_ would be maximal as well as the possible damage. Indeed, coincidence between mitochondrial aconitase inactivation and cell death has been observed [12]. However, when considering exposure to exogenous H_2_O_2_, our study indicated protection from H_2_O_2_ action of the mitochondrial aconitase. This highlights that the topology is inversed between mitochondrial ROS mediated damage and H_2_O_2_ treatment. Possible explanation of the relative immunity of the mitochondrial matrix with regard to exogenous H_2_O_2_ could be diffusion limitations [17] or/and large excess in antioxidant defense, the latter expected to be also efficient against mitochondrial H_2_O_2_. 

In conclusion, one may consider that mitochondria protect themselves and other cellular components from oxygen damage by three different means: avidly consuming oxygen, antioxidant defenses, and contribution to cellular bioenergetics to feed the renewal of damaged components. Mitochondrial impairment would in the first instance downgrade the above-mentioned protective roles. Further degradation would render mitochondria unable to check for their endogenous ROS and, worse, could increase their generation rate. Therefore, mitochondrial ROS release would not cause but rather highlight the decline in cellular bioenergetics and both would contribute cell death. 

## 4. Materials and Methods 

All cells were grown at 37 °C with 5% CO_2_ in media supplemented with 10% fetal bovine serum, 50 U/mL penicillin, and 50 µg/mL streptomycin. CHO-K1 cells (ATCC CCL-61) were grown in Ham’s F-12 medium containing 10mM glucose, 1mM glutamax, and 1mM sodium pyruvate (Gibco 31765-027). Human lymphoblasts (HPB-All, Jurkat, DND, Molt4) were obtained from Françoise Pflumio (CEA, Fontenay aux Roses France) and were grown in RPMI medium (Gibco 61870-010). Human immortalized fibroblast cell lines were established by one of the co-authors (AL) and grown in DMEM with 1mM Glutamax 5mM glucose. For oxygen flux measurements the cells were resuspended in their culture medium after action of trypsin 0.05 g/L in phosphate-buffered saline (PBS) containing 1 g/L EDTA. 

Oxygen concentration at 37 °C was monitored with an Oroboros “O2k” (http://www.oroboros.at/index.php?home). Oxygen consumption/production rate was calculated from oxygen concentration variation either with the DatLab software, which averaged rates over 80 s, or by calculation of individual values between two successive measurements (2 s) when fast rates had to be considered (early phase of H_2_O_2_ consumption and O_2_ release). H_2_O_2_ solution was prepared each day of experiment from stock solution (Sigma ref. H1009). The concentration was checked by UV absorbance using the extinction coefficient at 240 nm of 46.3 L × mol^−1^ × cm^−1^. 

To evaluate the impact of H_2_O_2_ on the aconitase to fumarase ratio H_2_O_2_ was added to CHO cells (approximately 2 × 10e6 cells per mL) incubated in the O2k 37 °C with stirring, and cell sampling was done 30 min after H_2_O_2_ addition. The aconitase and fumarase activities were measured in 96 well plates allowing measurement at 240 nm (Greiner Bio-One 675801) with a microplate reader equipped with two injectors (TECAN Infinite M200). Preparation of the plate was as follows: first 20 µL of detergent solution in PBS was introduced in the wells, followed by 20 µL of the cellular suspension in PBS, and, finally, by 50 µL of Tris Buffer 0.5 M pH 7.4. In our study 36 wells were used simultaneously (12 samples in triplicates), and optimization led to 100,000 CHO cells per well with final concentrations of 0.01% digitonin and 0.05% Triton X100 to probe for cytosolic enzymes or cytosolic and mitochondrial enzymes, respectively. The plate was introduced in the plate reader to start the reading protocol for measurement of OD240 nm: (1) Shaking (15 s, 2 mm linear); (2) Four readings (each 60 s) to evaluate the background and stability; (3) aconitase reaction with injection of isocitrate solution 120 mM (25 µL at 200 µL/s), shaking as before, twelve readings (each 40 s); (4) fumarase reaction with injection of malate 250 mM, rest of the procedure as before. Calculation of the aconitase to fumarase ratio was made with the assumption that the OD240 increase summed the rates of aconitase and fumarase reactions, which furthermore shared the same proportionality between OD increase and enzymatic reaction rate.

## Figures and Tables

**Figure 1 ijms-21-09149-f001:**
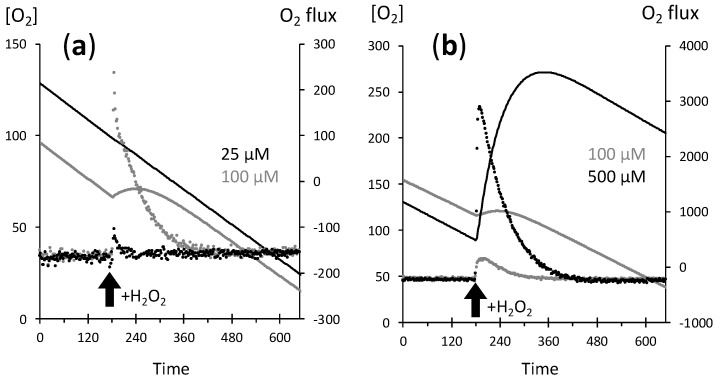
O_2_ release upon H_2_O_2_ addition. Chinese hamster ovary (CHO) cells suspension at approx. 2 × 10^6^ cells/mL; *X* axis = time in seconds; left *Y* axis = recorded oxygen concentration expressed in µM (lines); right Y axis = oxygen flux rate in pmol/(s.mL) calculated over a two second interval (dots); and thick upward black arrow = H_2_O_2_ addition performed at time 180 s. The stable negative value of oxygen rate before H_2_O_2_ addition is the oxygen consumption due to cellular respiration. Different H_2_O_2_ additions are represented: (**a**) 25 µM (black symbols) or 100 µM (grey symbols) H_2_O_2_ in the cellular suspension, (**b**) 500 µM (black symbols) or 100 µM (grey symbols) H_2_O_2_ in the cellular suspension.

**Figure 2 ijms-21-09149-f002:**
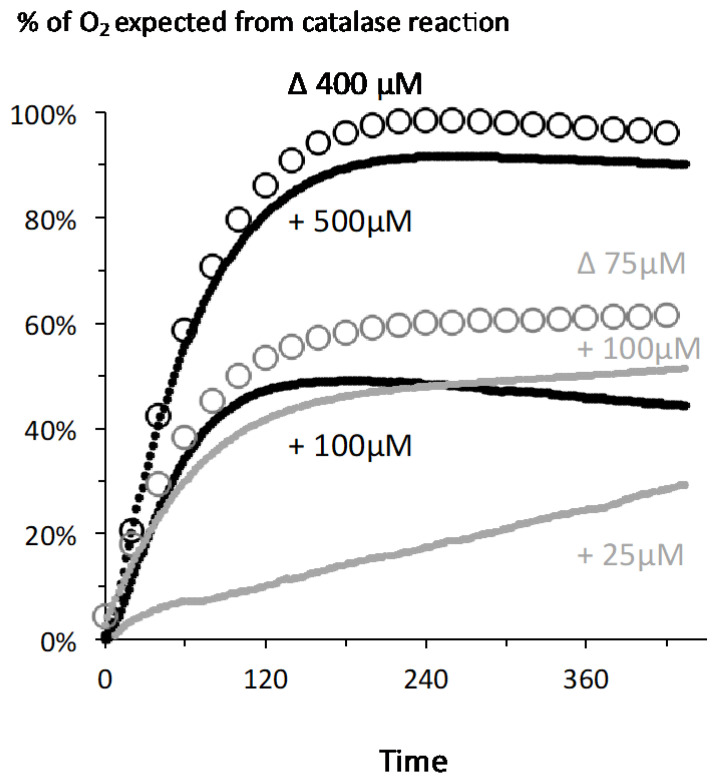
Increase in oxygen content in the medium compared to a full catalase reaction: *X* axis = time in seconds with addition of H_2_O_2_ considered as zero time; *Y* axis = oxygen release expressed as percentage of the amount expected from a 100% yield for the catalase reaction (O_2_ = ½ H_2_O_2_). The two experiments presented in Figure 2a are in black symbols (+100 and +500 µM H_2_O_2_) while those presented in Figure 2b are in grey (+25 µM and +100 µM). Empty symbols represent results obtained when subtracting the oxygen release in the two concomitant chambers, i.e., either 100 − 25 = 75 µM H_2_O_2_ or 500 − 100 = 400 µM, with one data point out of ten shown for the sake of clarity.

**Figure 3 ijms-21-09149-f003:**
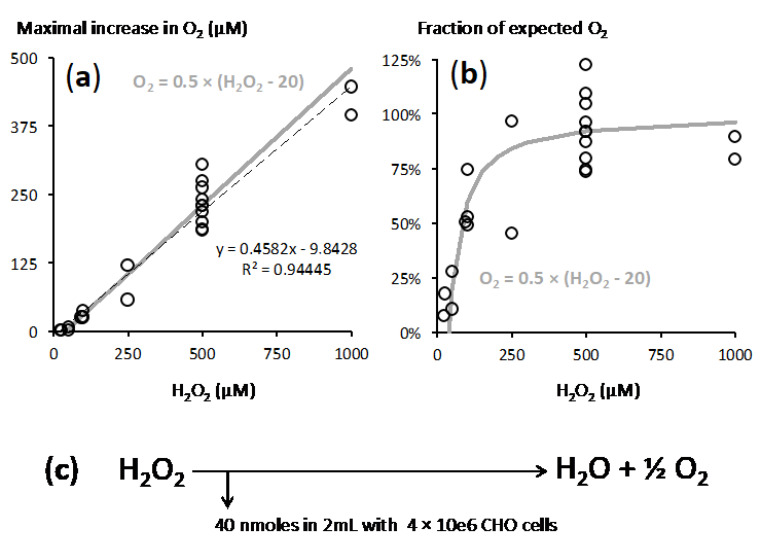
Recapitulation of experimental data led to a model for the reaction between H_2_O_2_ and CHO cells. CHO cells suspension ranged from 1.5 to 2.2 × 10^6^/mL. (**a**) *X* axis = final H_2_O_2_ concentration in the medium, *Y* axis = maximal observed increase in O_2_ (see Figure 2 for an example); dotted black line = linear fitting according to the equation shown in grey (with values in µM) in the panel upper part; grey lines = released oxygen given by the predicted equation of the model. (**b**) Expression of these O_2_ increases as percentage of the value expected from full engagement into catalase reaction, same X axis and symbols as in panel a. (**c**) Resulting model: upon H_2_O_2_ addition, 40 nanomoles H_2_O_2_ are immediately engaged in reactions not leading to O_2_ release; the rest of H_2_O_2_ is subjected to catalase action.

**Figure 4 ijms-21-09149-f004:**
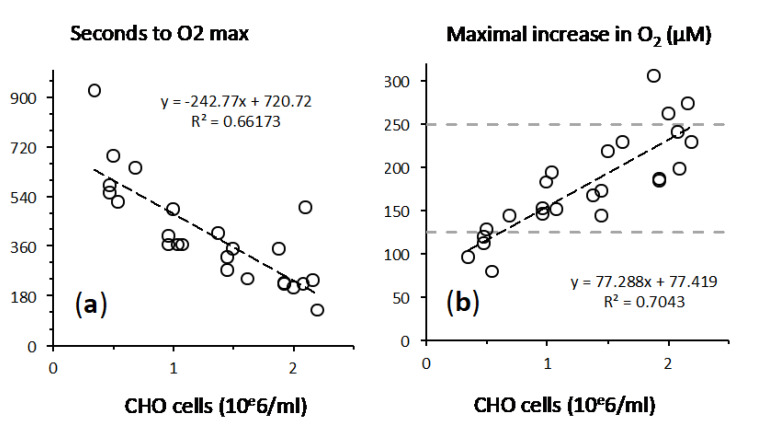
Influence of cell density on O_2_ release. *X* axis = concentration of CHO cells exposed to a single addition of 500 µM H_2_O_2_. (**a**) Time delay to reach the maximal O_2_ increase, inverse to the reaction rate. (**b**) O_2_ release in µM with dotted grey lines figuring the values corresponding to a 50% and 100% yield of O_2_ recovery from the catalase reaction (125 µM and 250 µM).

**Figure 5 ijms-21-09149-f005:**
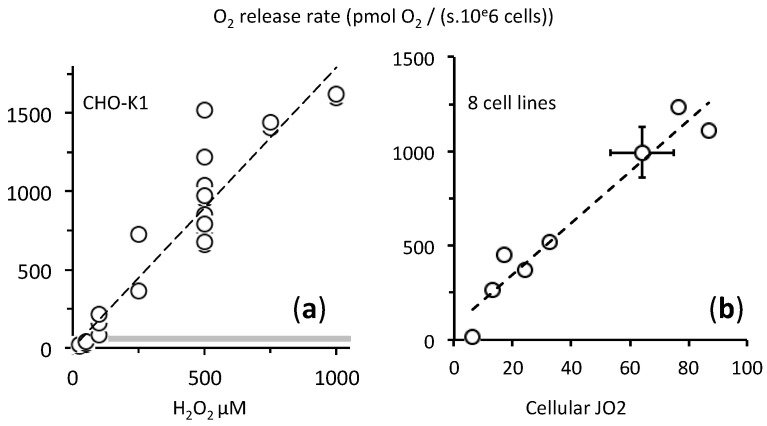
High initial rates of O_2_ reflect the catalytic efficacy of catalase reaction. *Y* axis = initial O_2_ release rate determined over 10 to 40 s after H_2_O_2_ injection; (**a**) *X* axis = H_2_O_2_ final concentration after injection; CHO-K1 cells suspension were in the range 1.5 to 2.2 × 10^6^ cells/mL; thick grey line = CHO-K1 cellular oxygen consumption (≈60 pmol O_2_/(second × 10^6^ cells) given for comparison’s sake; dotted line = linear fitting, which suggested the absence of saturation up to 1 mM H_2_O_2_. (**b**) The catalase initial rate after 500 µM H2O2 injection was determined with eight different cell lines. *X* axis = endogenous cellular oxygen consumption rate (respiration) before the single 500 µM H_2_O_2_ addition. The different cell types are listed according to increasing *X* values: human lymphoblasts (HPB-All, Jurkat, DND, Molt4), human immortalized skin fibroblast, CHO-K1 ± SD (*n* = 21 more data than in (**a**), cell density from 0.35 to 2 × 10^6^ cells/mL), osteosarcoma 143B, and human neuroblastoma (SH-SY5Y). The dotted line represents linear fitting with values *Y* = 14*x* + 75 with *R*^2^ = 0.94.

**Figure 6 ijms-21-09149-f006:**
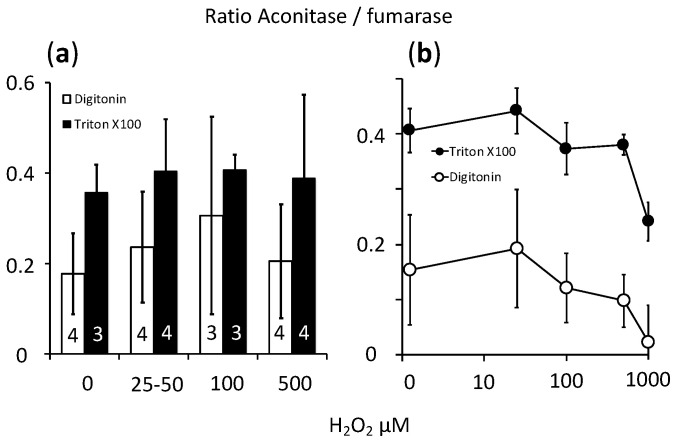
Evaluation of cellular oxidative stress using the aconitase to fumarase ratio in CHO cells exposed to H_2_O_2_. *Y* axis = aconitase to fumarase ratio; *X* axis = H_2_O_2_ concentration. (**a**) Empty bars = values observed in cells treated with digitonin (cytosolic activities); filled bars = values observed in cells treated with Triton X100 (both cytosolic and mitochondrial activities); values expressed as mean ± SD; the number of independent experiments is indicated at the bottom of the histogram bars. (**b**) Single experiment with the concentration of H_2_O_2_ raised to 1 mM; values expressed as mean and SD of the wells observed in that experiment.

**Figure 7 ijms-21-09149-f007:**
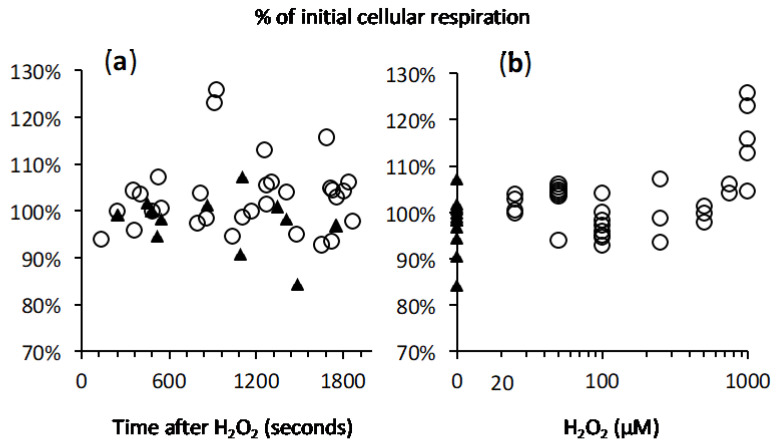
Absence of impact of H_2_O_2_ on CHO cells respiration. *Y* axis = cellular oxygen consumption after H_2_O_2_ had been exhausted and expressed as percent of the initial rate (rate recorded just before H_2_O_2_ addition). (**a**) *X* axis = lag time between H_2_O_2_ addition and measurement oxygen consumption rate; black triangles refer to concomitant control experiments (without H_2_O_2_). (**b**) *X* axis = amount of H_2_O_2_ injected in the experiments represented in panel a.

**Figure 8 ijms-21-09149-f008:**
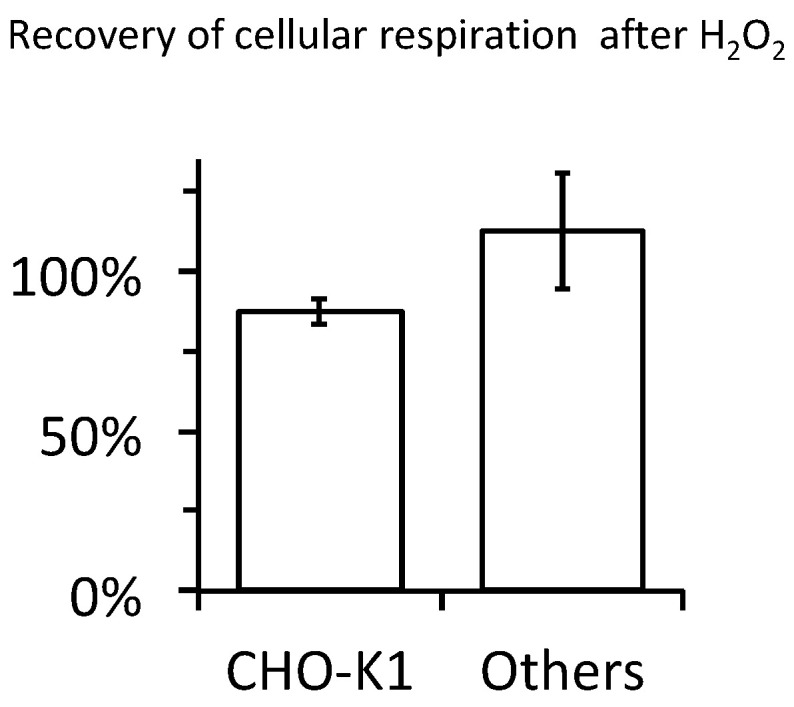
Absence of impact of H_2_O_2_ on the respiration is common to different cells. Oxygen consumption rate after H_2_O_2_ had been exhausted is expressed in % of the reference rate for cells exposed to 500 µM H_2_O_2_; values expressed as mean ± SD; CHO-K1: 21 preparations of CHO-K1 with cell density ranging from 0.35 to 2.08 × 10^e^6 cells per mL. Others: 7 preparations of the cell lines used in Figure 5b (one value per cell line cell and density ranging from 0.86 to 3 × 10^6^ cells per mL).

**Figure 9 ijms-21-09149-f009:**
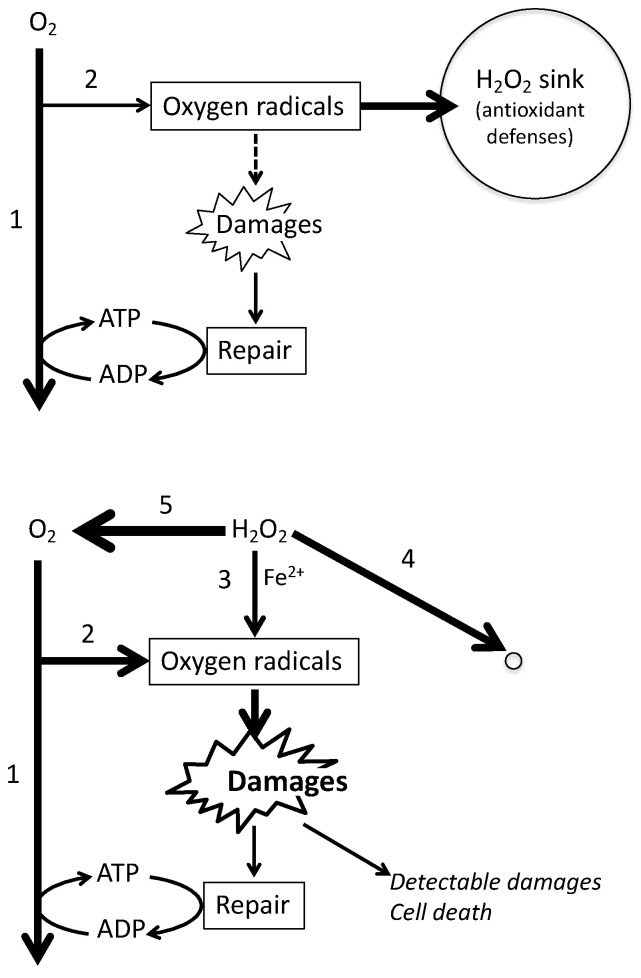
Scenario for H_2_O_2_ mediated oxidative damage. **Top:** normal cellular physiology, endogenous oxygen metabolism is essentially devoted to mitochondrial ATP generation (1). This and other oxygen using enzymes generate oxygen radicals (2) and a vast majority is eliminated by antioxidant defenses. Some oxidative damages (Damages) occur and cellular bioenergetics contributes to their reparation resulting in a steady state with no accumulation of oxidative lesions. **Bottom:** addition of H_2_O_2_ (high µM) has three immediate consequences: it increases the probability of Fenton’s reaction (3), H_2_O_2_ reacts within seconds with existing reactive molecules and annihilates the existing antioxidant defenses (4), and H_2_O_2_ concentration drops abruptly through action of cellular catalase. This increases greatly the intracellular O_2_ concentration (5) and hence endogenous oxygen radicals production (2). On one side oxygen radical generation is increased (2 and 3) and on the other side antioxidant defenses are invalidated (4). This is expected to generate oxidative damages at rates exceeding repair: they accumulate and become detectable and may result in cell death.

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
