# Peer review of "Use of H2O2 to Cause Oxidative Stress, the Catalase Issue"

_ijms, 2020, doi:10.3390/ijms21239149_

Round 1
Reviewer 1 Report
The manuscript by Ransy and colleagues use the polarographic measurement of oxygen concentration in cellular suspensions and demonstrates that H2O2 addition results in O2 release in cell culture. Although it is an interesting topic and in general the manuscript is well organized, the section "Methods" is not appropriately written.
Major points
The Methods section does not allow the reader to repeat the experiments what is crucial in scientific reports.
For example, what was the medium in which CHO cells were cultivated, how the cells were obtained etc.
Minor points
1) The manuscript needs language revision as there are some mistakes, like "hydrogen peroxyde" which is present throughout the manuscript.
2) Figure 1 is unnecessary. You can mention it in the text only.
3) List of abbreviations is almost anecdotic. Why are there given abbreviations like MDPI, DOAJ etc. and ROS and so on is not listed?
4) Other minor grammar and typos should be corrected. The manuscript would profit from a professional editing service.
Author Response
Thank you for your comments, you'll find hereafter our answers/modifications in response to the points raised:
Major point: Methods section
We have included a description of the origin of the cell lines as well as their culture conditions. We thought it was not necessary as we used the usual cell culture procedures.
Minor points
1 & 4) Langage: the manuscript has been amended to improve lisibility (a reader used to write scientific articles for decades has suggested modifications). In addition following demand from the other reviewer a conclusion scheme is now proposed (new Figure 9). We corrected as much as we could mistakes like "peroxyde". A comparison of the old/new version is included.
2) Figure 1 has bbeen replaced by a text description of the values, we propose to include the histogram in Appendix B.
3) List of abbreviations has been adapted to the present manuscript.
Reviewer 2 Report
The paper is well written and focuses on a relevant issue. It could be useful a figure summarizing main findings
Author Response
Thank you for your comments.
Following your suggestion we have now included a figure summarizing our conclusions (present Figure 9).
The manuscript has been amended (we hope improved) in the present version.
You will find in the accompanying document a comparison of the two versions.